# OpenReview forum: "Style Outweighs Substance: Failure Modes of LLM Judges in Alignment Benchmarking"
_ICLR.cc/2025/Conference — ICLR 2025 Poster_

### Official Review · Reviewer_ngWC · 2024-11-01

**Soundness:** 2
**Presentation:** 3
**Contribution:** 2
**Rating:** 6
**Confidence:** 4

**Summary:**

This paper investigates the reliability of benchmarks that employ LLM-as-a-judge evaluation of model alignment. Specifically, it aims to investigate:
1. Correlation of LLM-judge preferences with more traditional (input, output) benchmarks, where the ground truth is known
2. Implicit biases of LLM-as-a-judge preferences when evaluating pluralistic objectives, that the authors explicitly disentangle into [completeness, correctness, safety, conciseness, style]

The paper also presents SOS-Bench, an amalgamation of existing traditional benchmarks, which is partitioned in a way to evaluate the HHH principles.

**Strengths:**

The research question the authors tackle is timely and relevant, especially given the increase in recent works that evaluate alignment largely using LLM-as-a-judge.

The work investigates questions that are in the community's consciousness and provides experimental evidence and analyses.

The paper reads well and is interesting. The presentation interleaves empirical analysis with intuition and justifications.

**Weaknesses:**

One of the main focus of this work is empirical analysis of the reliability of LLM-as-a-judge benchmarks. As such, the main questions on my mind were whether the empirical findings were **robust** and **systematic**, and adequately controlled for potential confounding factors.

Below are my **major concerns**:
1. **LLM judges [S4]**: The authors evaluate results on one judge model (either gpt4 or gpt4o-mini) and one judge benchmark (Arena-Hard-Auto). I understand the effort and costs involved in evaluating multiple judges and benchmarks, but some consistency would strengthen the findings here. Nonetheless, I found this to be the most convincing set of results in the paper.
2. **Novelty of SOSBench [S5]**: Based on my understanding, SOS-Bench is an amalgamation of 19 existing benchmarks, binned together to evaluate three aspects of world knowledge, instruction following, and safety. I am not confident that this can be claimed as a novel contribution, especially since reporting evaluations on multiple benchmarks is common practice in alignment papers/ML research in general. If the author's contributions here are in collating and integrating them, I would consider it more as valuable engineering work rather than a research contribution.
3. **Implicit bias from LLM vs implicit bias from benchmark selection [S5]**: The authors could consider being more nuanced and mindful when discussing different approaches to evaluating alignment. Specifically, the principles of alignment are often pluralistic and ill-defined, e.g. 'safety' or 'helpfulness'. Distilling such principles into an explicitly specified metric or evaluating them indirectly (through traditional benchmarks) is also susceptible to its pathologies. For example, the authors mentioned `BLEU`, which has several well-documented biases. Additionally, the authors have introduced their own biases when using the performance of *world knowledge* as a surrogate for honesty, *instruction following* for helpfulness, and *safety* for harmlessness. While these might be reasonable surrogates, the point is, that the authors have injected their own implicit/explicit biases into the selection of benchmarks and thus evaluation metrics.
4. **Robustness of findings [S6]**: I have concerns about the robustness of the findings and the systematicity of experimental design. I will list my questions in detail below:
* `Data scaling improves alignment in the SFT stage of post-training.` The authors have controlled for differences in pretraining by starting from the same seed model. Is this paragraph (and Fig 2) only comparing models after SFT or after SFT + preference alignment? Analyzed by itself, the results appear to indicate that models are better aligned when SFT is performed with more data. This alone is not surprising. The authors also mention that SFT scale is more important than **curation** and **other important factors**. This relative impacts (of these other important factors) are not analyzed/reported, so it is unclear whether scale or a latent confounder has a higher impact on alignment post SFT.
* `Specialist post-training methods underperform generalist methods.` The authors mention 'diversity of prompts'---this would benefit from an explicit definition. Is it the narrow/broadness of the SFT dataset? Here, the comparison does not control for the size of the dataset, so it is not clear whether the observed differences in performance on specialist domains is due to 'scale' or 'prompt diversity'. Additionally, the authors have not investigated the quality of the datasets compared. I tried to have a look and couldn't find some of these datasets (e.g. Code LLAMA 3) online (the implication here is that they might be lower quality). In short, the results are not strongly convincing that 'diversity of prompts' is the primary factor that could explain the difference in downstream performance.
* `Preference optimization trades world knowledge for improved safety and instruction following.` This is a well-documented phenomenon colloquially referred to as the 'alignment tax'. Additionally, while the authors acknowledge that this result could be confounded by the choice of preference dataset, I feel that this as a crucial factor that must be controlled for any empirical insights to be made.
5. **SFT vs PO**: In the abstract, the authors claim `supervised finetuning stage, and not the PO stage, has the greatest impact on alignment'. Perhaps I missed it, but where this substantiated or investigated?

IMO, this work is ambitious (which I applaud), attempting to tackle many questions but unfortunatley compromising on thoroughness and depth. Specifically, the authors try to approach two tasks in one paper: (1) investigating pathologies of LLM-as-a-judge evaluations, and (2) proposing their own benchmark. I can see that the two are related. But my feeling is the impact would be significantly improved if the paper focused more in-depth on analyzing either of the two.

**Questions:**

I also have some minor clarifying questions and suggestions:

1. The paper is missing many in-text references (e.g. for the datasets in Tables 4 and 5), making it slightly challenging for the reader to form context when analyzing the presented results.
2. How is `Loss` defined in Table 3?
3. In Table 5, is Arena score based on an 'overall judgement'. If so, it would be interesting to also compare against fine-grained evaluations of [completeness, conciseness, style, safety, correctness] to see if there are any correlations there with WK, IF, S.
4. The authors should reference pluralistic alignment (which concerns a similar effect as those analyzed) in S4.1 and Table 2.
5. The authors mention the Bradley-Terry model somewhat ad hoc. Did the authors intend to imply anything about the BT model through the presented results/analyses?

---

> ### Author Response · Authors · 2024-11-22
>
> We thank the reviewer for a comprehensive and thoughtful review!
>
> **W1.** We agree with this note. We now report on four judges instead of one – please see our global response for more details.
>
> **W2.** We thank you for the note – we feel that SOS-Bench has the potential to be high impact and therefore of interest to the community; for more on why, please see our global response CA1. We also propose to make our technical contribution more novel. After evaluating the literature, we propose to use the method of tinyBenchmarks (ICML 24, https://arxiv.org/abs/2402.14992) to train and release robust IRT++ estimators for each of the datasets in our meta-benchmark; we estimate that this should allow for accurate forecasting of LLM performance with approximately two orders of magnitude fewer samples. Unfortunately, computational constraints prevent us from providing these estimators now, but we will do so before the camera-ready deadline.
>
> **W3.** We agree that all alignment benchmarks, including ours, introduce some biases; we discuss the limitations of our approach at some length in Section 9 (impact-limitations). However, if there is any particular language which concerns you, we would be happy to examine it.
>
> **W4.**
>
> 4a. Thanks for the question. We are comparing only SFT models in Fig. 3. While it may not surprise you, we think that the result that increased scale alone is the best predictor of model alignment would be of interest, since prior heavily cited literature, such as LIMA (https://arxiv.org/abs/2305.11206) and Phi (https://arxiv.org/abs/2306.11644) have argued that quality trumps quantity.
>
> 4b. Thanks for this note. Yes, we mean by this the breadth of domain knowledge and the variation in semantic structures among prompts. We have added this text to the paper.  While it is true that we have fewer datasets from which to choose here, we do have specialist datasets (such as FLAN and Replete Coder) which are nearly as large as our largest non-specialist datasets; therefore, we believe that scale has been, to an extent,  controlled for. As for the quality, all of the datasets we choose were curated by AI startups or prominent researchers, and were popular on HuggingFace, which is at least some testament to their quality. Incidentally, we noticed that the references for Code Llama 3, Tulu Human and Numina CoT were missing from Table 7. We apologize for this and we have added them.
>
> 4c. Our understanding from the literature was that while the alignment tax has been documented, that it remains controversial; therefore, we think the experiments are still helpful. We would be happy to add some relevant prior work citations if you have suggestions. As for controlling for the effects of dataset, SOS-Bench is a meta-benchmark of 19 datasets, and therefore already controls (to the extent possible) for the quirks of any one of them. We do agree, however, that it is valuable to ablate the choice of questions we present in our LLM judge results. Therefore, for this rebuttal, we curated a new set of 500 questions expressly for this purpose, and added the results of this experiment to the appendix – please see our global response for more details.
>
> **W5.** Thanks for the note – we agree that this claim was not scoped carefully enough. We have edited our abstract to read, “the supervised fine-tuning (SFT) stage of post-training has a large impact on alignment, with data scaling and prompt diversity as the driving factors”.
>
> **Q1.** We thank the reviewer for this note. For the camera ready draft, we will release all of the relevant checkpoints and reference the links in those tables (we cannot do so now because it would break anonymity).
>
> **Q2.** Thanks for this question – we define it as the percentage difference between the base score and the new score (we have added this to the paper).
>
> **Q3.** It is based on the overall score. We agree that it might be interesting to see how the behavior changes factor-wise, however, this would be somewhat expensive to test, so we will leave it to future work.
>
> **Q4.** Thanks! We have added a reference to pluralistic alignment to our related work section.
>
> **Q5.** We did not intend to imply anything beyond what we wrote, no.
>
> If you now feel more positive about our work, please consider raising your score. If you have more questions or concerns, please feel free to let us know.

---

> > ### Comment · Reviewer_ngWC · 2024-11-26
> > **Response to Rebuttal**
> >
> > I thank the authors for their hard work and commitment to address reviewer concerns.
> >
> > **W1.** Thank you for conducting this experiment, my concern here has been addressed.
> >
> > **W2.** I see, releasing IRT estimators does sound like a good idea, and would enhance the impact/utility of SOSBench. It is a bit hard for me to evaluate this contribution, since without experimental verification, it is hard to judge whether the IRT estimators are well-correlated with evaluations conducted on the full benchmark. On the other hand, I also recognize that training these estimators is impractical given the rebuttal window.
> >
> > **W3.** Thanks, I think many, if not all, evaluation of alignment will necessitate some bias. It is good to see these being openly acknowledged.
> >
> > **W4.** My additional suggestion for the authors is, before diving into the results, explicitly list all the plausible factors that could affect post-alignment/SFT evaluations. For example, for `W4a`, it would be easier to see explicitly that (1) quantity, and (2) quality are the two main factors being investigated, followed by a formal/clear definition of what these factors are. For `W4b`, it might be (1) scale, (2) diversity of prompts, (3) quality etc. In a way, by listing the hypothesis before experiments/results, it would be much easier to parse and help the readers form their own opinions on whether all sources of confounding have been controlled. Can the authors provide a table tabulating these hypothetical factors and how they have been controlled? Something brief would suffice, I trust they will be reflected in the paper in the future.
> >
> > Thanks.
> >
> > The reviewer

---

> > > ### Author Response · Authors · 2024-11-27
> > >
> > > We thank the reviewer for a quick and thoughtful response!
> > >
> > > **W1, W2, W3.** Acknowledged, thank you!
> > >
> > > **W4.** Here is the table and some discussion of the topics you raised. We are happy to add it to the camera ready draft of the paper.
> > >
> > > Our working hypothesis in the section was that a naive baseline ("curation method doesn't matter and we can predict outcomes by scale alone") would fit our data well when using grounded benchmarks (SOS-Bench), and less well when using LLM Judges. If true, this finding would support our claim that it is important to use other (non LLM-judge) measures of progress in alignment to supplement LLM judges.
> > >
> > > Our general approach to handling confounds was as follows. We introduced models representing various potential confounds. If the confound was impactful, then the relevant points would appear as strong positive outliers on the R, G, B lines in Fig. 3. In fact, we saw only strong negative outliers, suggesting that none of these confounds had a significant impact.
> > >
> > > Fig. 3: list of confounds and control models.
> > >
> > > NOTE: We measure semantic diversity by estimating the number of unique tokens in a sample of prompts and responses and heuristically setting a cutoff.
> > >
> > > | Confound Name | Relevant Models |
> > > |----|----|
> > > | Human authored prompts | Wildchat, ShareGPT, FLAN, Tulu-Human, LIMA |
> > > | Human authored responses | FLAN, Tulu-Human, LIMA |
> > > | Synthetic GPT responses | Magpie |
> > > | Technical subject matter focus | Code Llama 3, Numina-CoT, MetaMath, Replete Coder, WizardLM |
> > > | Validated correct answers | FLAN, Tulu-Human, Numina-CoT, MetaMath |
> > > | Semantically diverse prompts only | Numina-CoT, MetaMath |
> > > | Semantically diverse responses only | FLAN |
> > >
> > > Tab. 4: Because of the relative dearth of models available, we were only able to control for scale in this table; however, we have no reason to believe that the factors which we established in Fig. 3 did not have a meaningful impact would have one here, as it is the same set of models and a subset of the same benchmarks. We are happy to add a comment to this effect in our paper.
> > >
> > > Tab. 5: This table compares 2-stage post-training to 1-stage post-training while controlling for dataset size (see column DS Size) and the choice of SFT dataset (we ablate over Tulu, Magpie and Ultrachat). In Appendix Table 9, we ablate the choice of post-training algorithm, and find the choice has a small but statistically signficant effect on non-LLM-judge benchmarks. One thing we do not ablate here is the choice of post-training dataset; we leave that to future work. The comments of Tab. 4 apply here as well.

---

> > > > ### Comment · Reviewer_ngWC · 2024-12-02
> > > > **Thank you**
> > > >
> > > > I thank the authors for their response, the latest round of which has addressed all my main concerns.
> > > >
> > > > Highlighting the confounds systematically is very helpful to me, as a reader and a reviewer. I still have some concerns about the robustness and novelty of the findings (as I mentioned before, some insights have been identified in previous works, and some of the confounds are not fully controlled for, despite the author's best efforts). But nonetheless, the empirical alignment evaluation is sufficiently thorough, and IMO augments existing findings in the field.
> > > >
> > > > Correspondingly, I have updated my rating to 6.

---

> > > > > ### Author Response · Authors · 2024-12-03
> > > > >
> > > > > We thank you for your comments and feedback, and for helping us improve our work.

---

### Official Review · Reviewer_8QfX · 2024-11-01

**Soundness:** 3
**Presentation:** 2
**Contribution:** 3
**Rating:** 6
**Confidence:** 4

**Summary:**

This paper studies the drawbacks of the LLM-as-a-judge framework and proposes SOS-Bench, a ground-truth meta-benchmark, to circumvent those drawbacks.

**Strengths:**

The paper proposes studying the drawbacks of the LLM-as-a-judge framework, a critical topic.
- The most interesting findings, in my view, are in Table 3;
- The approach to studying biases reported in Table 3 seems sound.

**Weaknesses:**

The paper's drawbacks are greater than its contributions. In the following, I list what I think the be the biggest weaknesses of the paper and offer some suggestions on some of these points:

1) The paper does not provide convincing evidence of biases of the LLM-as-a-judge framework except from what is presented in Table 3. I believe the results in Table 2 are misinterpreted as they do not provide enough evidence of such biases (e.g., style over content preferences). I have two main reasons to support my claim:
     (i) The rank correlation of content and overall score in Table 2 seems to be very high. The fact that the correctness scores are "compressed" does not mean anything by itself. For example, that fact could happen if the benchmark has a good number of very easy or hard questions;
     (ii) "Completeness" seems to be more related to content/correctness than to style.

     Suggestion 1: for Table 2, it would be informative to present the rank (and Pearson) correlations between overall scores and specific scores.

     Suggestion 2: you could try regressing the overall score against the fine-grained scores and statistically test for the coefficients of this regression model. One limitation is that your covariates can be highly correlated to each other, inflating the estimate variances; you might need to collect more data. A second idea is running factor analysis in the full set of scores and then analyzing the low-dimensional factors.

     Suggestion 3: it would be more interesting to focus more on controlled experiments as those presented in Table 3.

2) I do not see a big value in introducing another meta-benchmark like SOS-Bench. We already have HELM and Open LLM Leaderboard, which are already very comprehensive. Moreover, size should not matter that much in the quality of benchmarks as emphasized by the authors; the accuracy of scores does not change much after a certain sample size (think about concentration inequalities, the law of large numbers, etc). Bigger sizes only make evaluation harder.

     Suggestion: instead of just combining many benchmarks by running LLMs in all of them, the authors could try to propose a method to derive a small and comprehensive "summary" (meta) benchmark. This summary benchmark could be much smaller (perhaps even ~100x smaller) than the current version of SOS-Bench and contain key/relevant examples/questions from each one of the three criteria in HHH. That would make the benchmark more usable and the contribution would be more expressive.

Minor points:
(i) Table 1 could be a correlation matrix.

**Questions:**

NA

---

> ### Author Response · Authors · 2024-11-22
>
> We thank the reviewer for a comprehensive and thoughtful review. We are glad that you appreciated the experiments and the approach to studying biases in Table 3.
>
> **W1.** We thank the reviewer for these suggestions. Following the line of your thoughts, we have revised our manuscript as follows (please see our global response for more details):
> 1. We now include the rank (and Pearson) correlations between overall scores and specific scores.
> 2. We now report on four judges instead of one.
> 3. We have added a factor analysis on the full set of LLM judgments and interpreted the results.
>
> **W2.** We thank the reviewer for the note. While we think that curating and retaining large benchmarks like SOS-Bench is valuable because it reduces the risk of contamination, we agree with you that a smaller summary benchmark is an important contribution in its own right. After evaluating the literature, we propose to use the method of tinyBenchmarks (ICML 24, https://arxiv.org/abs/2402.14992) to train and release robust IRT++ estimators for each of the datasets in our meta-benchmark; we estimate that this should allow for accurate forecasting of LLM performance with approximately two orders of magnitude fewer samples. Unfortunately, computational constraints prevent us from providing these estimators now, but we will do so before the camera-ready deadline.
>
> Regarding your question about why it is necessary to introduce another meta-benchmark beyond HELM and the OpenLLM leaderboard, please see our global response **CA1**.
>
> If you now feel more positive about our work, please consider raising your score. If you have more questions or concerns, please feel free to let us know.

---

> > ### Comment · Reviewer_8QfX · 2024-11-24
> >
> > Thank you for your rebuttal! I appreciate your effort. I will keep my score for now based on the observations below.
> >
> > **W1** I think it is still hard to disentangle style from the overall quality of the text generation. For example, we see that correctness and style heavily load on the same factors in your factor analysis. For this reason, I am not sure you can claim that "style outweighs content" based on the analysis in Figure 2/Table 2 (even though that might be true). Also, I insist that a correlation matrix of all factors shown in Fig2/Tab2 would be very helpful for us to understand how these things correlate with each other. I think it's going to be hard to come up with a convincing analysis if you do not focus on controlled experiments like the one shown in Table 3.
> >
> > **W2** I am not sure the ideas introduced in tinyBenchmarks would be well-suited for you here. I don't think it is interesting to only summarize existing benchmarks (like they did in their paper). In my view, the most interesting aspect here is combining existing benchmarks in a way that makes sense for your final goal. The final SOS-Bench would be a curated version of the existing one.

---

> > > ### Author Response · Authors · 2024-11-25
> > >
> > > We thank the reviewer for engaging with our work during this discussion period, and appreciate the additional feedback! We respond to the remaining criticisms below.
> > >
> > > **W1**. Our claim is that LLM-judges for alignment benchmarking have powerful implicit biases, and specifically, they prioritize stylistic preferences over other important considerations, like factuality and safety. The evidence supporting this finding is quite robust. The analysis in Figure 2 / Table 2, now extended to four judges, supports it. Table 3 provides further evidence. The R2 scores for each factor we have added also support it; the style-overall correlation is literally perfect. Even the factor analysis supports the claim; while there are indeed many cases where style and correctness are deemed synonymous by the judge (accounting for around 40% of the overall variance), the second largest factor loads principally on correctness, accounting for 12%, and a third factor, also stylistic, "brevity", accounts for another 12%.
> > >
> > > While the case is far from closed, and future work is free to argue the opposite, our observations are novel and would be of interest to the community. Still, perhaps it would help to word the claim a bit more cautiously? Something like, "Our claim is that LLM-judges for alignment benchmarking *can* exhibit powerful implicit biases, *such as* prioritizing stylistic preferences over other important considerations, like factuality and safety."
> > >
> > > **W2.** It's not entirely clear how to reconcile this observation with your prior one. Your suggestion was to derive a small and comprehensive "summary" benchmark, perhaps ~100x smaller. Using the TinyBenchmarks method on each of the tasks in SOS-Bench (independently), we expect to be able to do just that; achieve the same experimental results as we attain in our main paper. In addition, the smaller benchmark would allow us to establish an online leaderboard for alignment methods. So it will be a curated version of the existing SOS-Bench.

---

> > > > ### Comment · Reviewer_8QfX · 2024-11-28
> > > >
> > > > Thank you for your response! I apologize for the delayed response, but fortunately, we will have extra days for the discussion.
> > > >
> > > > **W1:** I appreciate your response. I think you have a point here and now I agree that Figure 3 provides good evidence of the alignment of style and overall score. I have some follow-up questions:
> > > > - How do you construct Table 2 considering multiple judges? It's not very clear from the text;
> > > > - Why do you get style scores, correctness scores, etc all with the same prompt? What are the advantages of doing everything together and separately?
> > > > - I realized your sample size is pretty big (when running factor analysis, for example). Have you considered my previous suggestion of running a regression analysis, regressing overall scores against the style, correctness, etc? I think this can be more convincing than just presenting a table or figure with marginal correlations. The advantage of regression is that it allows you to *control for other factors*; that is, you could respond to questions like: "What would happen to the overall score if I fix style but only vary correctness?" If you are using Python, the package Statsmodels offers an easy way to run this kind of analysis and lets you retrieve some diagnostics and confidence intervals for estimates. If your data is ordinal (which seems to be the case) you can consider using ordinal regression. If you have multiple judges, I would run this separately for each judge since they can have different biases.
> > > > - The same is true for factor analysis (regarding ordinal data); if you think it is more appropriate, you could try running polytomous factor analysis in a future version of the paper if that makes sense to you (for now, this is not so important in my view, and wouldn't affect my ratings).
> > > > - I see you used the varimax rotation; sometimes it's hard to interpret this rotation because it requires factors to be uncorrelated. You could also explore other oblique rotations (e.g., geomin) and hopefully, you're able to extract more insights. Also, this is a point to consider for a future version of the paper and you do not need to worry about this now.
> > > >
> > > > **W2:** Maybe I was not clear about this point; I am sorry about the confusion. What I meant from my comment is that tinyBenchmarks will only summarize each benchmark separately but it is not guaranteed to focus on what is really important for you. For example, if you are using benchmark X, there might be a set of examples in that benchmark that are not relevant to you at all (e.g., they might not reflect alignment). In that case, you do not want to include these questions but tinyBenchmarks would select some of them. The way I see this is that you could follow a two-step procedure:
> > > > 1. For each benchmark, you can first filter questions that are not important for you following some criteria. That would be the curation step;
> > > > 2. You use tinyBenchmarks to obtain a summary for each one of the remaining questions in each benchmark.
> > > >
> > > > Do you think this makes sense? Please let's discuss more this!

---

> > > > > ### Author Response · Authors · 2024-11-28
> > > > >
> > > > > Thanks for the reply; it is indeed good that we have more time now in the rebuttal period. We are glad you agree that Figure 3 provides good evidence of the alignment of style and overall score!
> > > > >
> > > > > **W1.Q1**: For each column, we report the average values over the panel of four judges. We can add this to the text before camera-ready.
> > > > >
> > > > > **W1.Q2**: One reason that we get all scores in the same response is because to get them in separate responses would result in an $n$-fold increase in the cost of the experiment, with $n$ equal to the number of factors (remember that each Arena-Hard-Auto run costs a few hundred dollars, depending on the judge and the number of models we want to compare). Another, more fundamental reason is that we are interested in whether LLM judges implicitly reweight explicit judgment criteria, and it is more natural to study this in a setting where we acquire all scores simultaneously.
> > > > >
> > > > > **W1.Q3**: Because the style score and overall score are identical, regression analysis failed to converge on all settings we tried in the past; however, we fulfilled your request now by randomly perturbing a few values of the style score, and by running regression analysis with style excluded.
> > > > >
> > > > > REGRESSION ANALYSIS (PERTURBED STYLE SCORE)
> > > > >
> > > > > Iterations: 260
> > > > >
> > > > > Log-Likelihood: -25.342
> > > > >
> > > > > | Variable | Coefficient | Std Error | z | P>z | 0.025 | 0.975 |
> > > > > |----------|------------|-----------|---|-------|--------|--------|
> > > > > | correctness_score | -1.3891 | 1.031 | -1.348 | 0.178 | -3.409 | 0.631 |
> > > > > | safety_score | 0.4807 | 2.184 | 0.220 | 0.826 | -3.799 | 4.761 |
> > > > > | completeness_score | 1.8916 | 1.041 | 1.817 | 0.069 | -0.149 | 3.932 |
> > > > > | conciseness_score | 0.1452 | 0.920 | 0.158 | 0.875 | -1.659 | 1.949 |
> > > > > | style_score | 24.0587 | 13.349 | 1.802 | 0.072 | -2.106 | 50.223 |
> > > > >
> > > > > * We see an overwhelmingly high loading of style score.
> > > > > * This result reflects the high degree of collinearity between Style and Overall.
> > > > >
> > > > > REGRESSION ANALYSIS (STYLE EXCLUDED)
> > > > >
> > > > > Iterations: 15
> > > > >
> > > > > Log-Likelihood: -6573.5
> > > > >
> > > > > | Variable | Coefficient | Std Error | z | P>z | 0.025 | 0.975 |
> > > > > |----------|------------|-----------|---|-------|--------|--------|
> > > > > | correctness_score | 1.3544 | 0.047 | 29.002 | 0.000 | 1.263 | 1.446 |
> > > > > | safety_score | -0.0721 | 0.108 | -0.668 | 0.504 | -0.284 | 0.139 |
> > > > > | completeness_score | 2.6563 | 0.050 | 53.392 | 0.000 | 2.559 | 2.754 |
> > > > > | conciseness_score | -0.1216 | 0.029 | -4.200 | 0.000 | -0.178 | -0.065 |
> > > > >
> > > > > * Here, the log-likelihood is much greater magnitude, and in fewer iterations, suggesting a stronger model capturing more variation in the data. Completeness score is strongly loaded, correctness somewhat more weakly, and conciseness has a negative loading with low standard error. Safety appears to have no significant impact.
> > > > > * For each one-unit increase in completeness, the odds of getting a higher final score increase by about 14.2 times, compared to 3.9 times for correctness. The far higher loading of completeness indicates that answers judged to be more complete and less correct have a high probability of beating answers judged to be less complete but more correct.
> > > > > * Since completeness and conciseness can be interpreted semantically as constituents of style (at least in part), the LLM judge is also implicitly reweighting explicit judgment criteria *for other judgment criteria*, which is interesting as well.
> > > > >
> > > > > **W1.Q4**: Because it was a very quick experiment to run, we ran factor analysis using `promax`, an oblique rotation method. The results for factor 1 were largely similar, with factor 2 and 3 somewhat differently weighted. This analysis accounted for less of the variance than `varimax` overall.
> > > > >
> > > > > ```
> > > > > Factor Loadings:
> > > > >                      Factor1   Factor2   Factor3
> > > > > correctness_score   0.811395  0.274827  0.279021
> > > > > safety_score        0.013918 -0.018421  0.191941
> > > > > completeness_score  0.924696 -0.110041 -0.003913
> > > > > conciseness_score  -0.016666  0.600223 -0.025163
> > > > > style_score         0.890143 -0.136471 -0.018206
> > > > >
> > > > > Variance Explained:
> > > > >                  Factor1   Factor2   Factor3
> > > > > SS Loadings     2.306252  0.466870  0.115674
> > > > > Proportion Var  0.461250  0.093374  0.023135
> > > > > Cumulative Var  0.461250  0.554624  0.577759
> > > > > ```
> > > > >
> > > > > **W2**: We thank you for the clarification. The idea of prefiltering responses before we use TinyBenchmarks is interesting. However, we are not aware of any algorithmic method which, with high confidence, could choose samples more related to the topic of alignment.  Even if one existed, or if we used human auditors to prefilter responses, it could raise concerns that we were cherry picking samples to support a particular hypothesis. We did carefully select tasks that are well established in the literature, representing a diverse set of problems in safety, world knowledge and instruction following, so we believe that the questions should be closely linked to progress in alignment.
> > > > >
> > > > > We hope that we have resolved your concerns. In any case, we are happy to keep the conversation going!

---

> > > > > > ### Comment · Reviewer_8QfX · 2024-12-02
> > > > > >
> > > > > > Dear authors,
> > > > > >
> > > > > > I appreciate your detailed responses and hard work. I did not realize that the style scores were **exactly** the same as the overall quality score. In that case, you could emphasize that and I would say the regression analysis does not seem needed. I have increased my score since you have a good contribution in terms of judges' biases. I am not still 100% convinced that SOS-Bench is a very expressive contribution the way it is, as I mentioned before; that's the reason why I will not increase the score to 8.

---

> > > > > > > ### Author Response · Authors · 2024-12-03
> > > > > > >
> > > > > > > Thanks for the response! We will emphasize the exact similarity in a future draft.

---

### Official Review · Reviewer_J78n · 2024-11-03

**Soundness:** 4
**Presentation:** 4
**Contribution:** 3
**Rating:** 5
**Confidence:** 4

**Summary:**

This work exposes the issue of significant implicit bias in benchmarks that employ LLM judges to evaluate the alignment with human preferences. They evaluate LLMs with concrete metrics and suggest that the post-training and evaluation for the alignment of LLMs should be further designed to focus on more specific desirable factors.

**Strengths:**

1. This work tested the bias of LLM-judge between and within criteria and provided convincing results.
2. This work analyzed the performance of the latest LLMs on concrete evaluation metrics and summarized the relationship between performance and post-training settings.

**Weaknesses:**

1. The SOS bench proposed in this paper is basically a combination of existing static evaluation datasets regarding instruction following, world knowledge, safety, etc. What specific limitations of existing static datasets does SOS-Bench address? Are there any novel data processing, aggregation, or evaluation techniques used in creating SOS-Bench?

**Questions:**

Please respond to the concerns in the "Weaknesses" part.

Q1. What specific role do you envision for static benchmarks with ground truth answers versus LLM-judge benchmarks in evaluating alignment? Are you suggesting completely replacing LLM-judge benchmarks or proposing a hybrid approach? How do you see these different evaluation methods complementing each other?

Q2. Could you provide more details on the process of converting LLM-judge preference results to the absolute percentage scores shown in Table 2? Specifically:

- 2.1 What exact steps were used to convert preference judgments to percentage scores?
- 2.2 Were any normalization or scaling steps applied?
- 2.3 How did you account for or mitigate potential biases in this conversion process?
- 2.4 Did you perform any sensitivity analysis on different scoring methods?"

---

> ### Author Response · Authors · 2024-11-22
>
> We thank the reviewer for a comprehensive and thoughtful review. We are glad that you find our results convincing, and we agree that it is important to supplement LLM judgments with concrete, interpretable evaluation metrics such as SOS-Bench.
>
> **W1.** This is a great question, thanks for asking it!
>
> LIMITATIONS OF EXISTING STATIC DATASETS: While existing premier static benchmarks such as HELM Core (itself a meta-benchmark) and LiveBench are very important and inspired aspects of our work, they are not optimally configured to be drop-in supplements for LLM Judges in the context of model post-training. LiveBench has an implicit dependence on instruction following because all of its questions are open-ended, making it difficult to precisely assess the performance of smaller and older LLMs which tend to be used in academic papers. The HELM family of benchmarks is complex, with many non-intersecting metrics, models and datasets; important factors in model alignment, such as safety (bias and toxicity in HELM) are not covered for all models, and instruction following is mostly evaluated implicitly. Finally, it is not obvious how one would  compare alignment methods (E.G., DPO vs PPO) using HELM.
>
> NOVEL TECHNIQUES USED IN SOS-BENCH: To supplement the technical novelty of SOS-Bench and improve efficiency, we propose to use the method introduced in tinyBenchmarks (ICML 24, https://arxiv.org/abs/2402.14992) to train and release robust IRT++ estimators for each of the datasets in our meta-benchmark and release them by the camera-ready deadline. We estimate that this should allow for accurate forecasting of LLM performance with approximately two orders of magnitude fewer samples. Beyond that, we believe that the novel combination of evaluations in SOS-Bench will be, as a unit, high-impact and worth sharing in the community because they will make it much easier to measure progress on salient factors in alignment, just as HELM Core (also a meta-benchmark) did for foundation models. Towards this goal, we provide code to reproduce our benchmark exactly (and we will provide all relevant model weights once we are de-anonymized), which will make it easy to compare future work to our baselines. We also note that high impact meta-benchmarks have served as a key contribution for multiple recent publications (ICLR24, ICLR24).
>
> **Q1.** Good question. We envision SOS-Bench and other static benchmarks serving as an important supplement to LLM judgments. Much as inter-rater agreement is considered an important measure of reproducibility in many fields, we believe that progress in alignment will ultimately be best measured via a small cluster of benchmarks utilizing diverse methodologies.
>
> **Q2.** We have added the following explanatory text to Page 3 of our revised manuscript, we hope that this answers your questions.
>
> The chosen LLM judge conducts a pairwise comparison against a baseline model (GPT-4-0314 in the original paper), scoring outputs on 5-point Likert scale. The judge generates its own response to the question before judging, and uses chain-of-thought prompting for consistent judgments. The paper implements a two-game setup to avoid position bias, aggregating 1000 judgments per model using Bradley-Terry, resulting in final scores and confidence intervals through bootstrapping. This judgment pipeline has been shown to have strong correlation with the judgments of humans. 95% confidence intervals are reported as part of the benchmark; these can be as high as 4% for judgments close to the 50th percentile score. However, we believe that these estimates are still too conservative; we instead report the variance induced by a series of ablation studies (see our revised Appendix, Section I.3).
>
> If you now feel more positive about our work, please consider raising your score. If you have more questions or concerns, please feel free to let us know.

---

### Official Review · Reviewer_NeWJ · 2024-11-04

**Soundness:** 3
**Presentation:** 3
**Contribution:** 3
**Rating:** 8
**Confidence:** 3

**Summary:**

This work investigates when LLM-judge preferences translate to progress on other metrics for alignment, and in cases of failure, why not. In particular, this work finds that LLM-judges have implicit biases, and prioritize stylistic preferences over eg factuality and safety. This work also introduces SOS-bench, a new alignment benchmark with ground truth and with the goal of gauging progress on helpful, honest, and harmless principles. Experiments show that data scaling in SFT and improving prompt diversity are important predictors of improved alignment.

**Strengths:**

1. The paper is well motivated and definitely investigates an important question. It is valuable to know the failure modes of LLM-as-judges.
2. Paper is well written and logically organized.

**Weaknesses:**

1. Since SOS-bench is a collection of existing benchmarks, I’m not sure whether this is a novel contribution (but rather a thoughtful choice for what benchmarks to use).
2. In table 2: Why not advocate for using correctness for the score (instead of using the raw score)?

**Questions:**

Figure 2 claims that “in the SFT stage of post-training, the size of the dataset, rather than the method used to curate the data, is the strongest predictor of alignment.” Does this assume that all the data is deduplicated? Also, prior works seem to indicate that higher quality data (and data from different domains) can be more important than size. Can the authors comment on this?

---

> ### Author Response · Authors · 2024-11-22
>
> We thank the reviewer for a detailed and thoughtful review! We are glad that you found our work well motivated and well organized, and agree that the questions we aim to address in this work are very important.
>
> **W1.** We thank the reviewer for this note. Please refer to our global response for our reply.
>
> **W2.** This is an excellent question. One reason we do not advocate for this is that correctness is not the only important factor in alignment. In certain cases, safety might take precedence over everything else; in others, fluency of style might be the main desiderata. Most often, end users are likely to want some bespoke blend of these criteria. Another reason is that even if correctness was the only consideration, models would likely exhibit implicit bias w.r.t. their weighting of violations (evidence in Table 3). We are happy to add this discussion point to our paper.
>
> **Q1.** Thanks for this question! We do not assume that all of the data is deduplicated, making our findings perhaps a bit more surprising. As you note, certain prior works (most notably LIMA: Less is More for Alignment (https://arxiv.org/abs/2305.11206)) have argued that quality is more important than quantity in the SFT stage of post-training. We evaluate models aligned using LIMA data alone and find that they perform towards the bottom of the range, as predicted from the scale of the data. As for data from different domains, we find that a diverse set of knowledge domains and a modest amount of data during post-training outperforms a large amount of data in one narrow knowledge domain, at least from an alignment standpoint (see Table 4).

---

> > ### Comment · Reviewer_NeWJ · 2024-12-02
> >
> > Thank you to the authors for their reply. I will maintain my score.

---

> > > ### Author Response · Authors · 2024-12-03
> > >
> > > We thank you for your comments.

---

### Author Response · Authors · 2024-11-22

We thank all the reviewers for their time and for providing constructive comments for enhancing the paper. We appreciate that reviewers recognized:

* Our work is well-motivated and well organized
* We provide convincing evidence of certain failure modes of LLM judges
* We provide important analysis on the performance of the latest LLMs using concrete evaluation metrics

We have conducted extensive additional experiments and added several new discussion points, which we hope will address some reviewer concerns.
We would be very happy to keep the discussion going, addressing any points that remain unclear, or any new suggestions. Thanks again for your suggestions!

## Revised Draft

Following helpful suggestions from our reviewers, we have added several new experimental results to the paper.
1. **(Improved presentation of main results.)** We replaced the table with a figure which is easier to analyze at a glance (we will still include the tables in our repository artifacts). In the figure, we now report the Pearson correlation for each factor w.r.t. overall score. We see that style score and overall score are perfectly correlated.
2. **(Ablating the choice of LLM judge.)** At a substantial cost of over $500 USD, we have followed the request of several reviewers by testing three additional judge LLMs; Claude Sonnet, GPT 4o, and GPT-3.5. We will include the raw results from these judges in our repository artifacts, and we report the aggregate correlations (Pearson and Spearman) in our main paper, Table 2. Remarkably, we find that across all four models, the average correlation between style score and overall score is still perfect.
3. **(Ablating the LLM Judge Pipeline.)** Some reviewers had questions about how much impact other potential confounds in the pipeline could be expected to have. We add 6 ablation studies, covering a wide range of variations, such as changing the semantics of the judge template, eliminating order-independent judging, changing the baseline model, and instructing the LLM judge not to give its own answer before judging; see Appendix I of our revised manuscript.
4. **(Ablating the choice of questions.)** In order to address concerns about how much the choice of questions in Arena-Hard-Auto impacts judgments, we run the same experiment again, substituting 500 questions from popular subreddits such as AskHistorians for the Arena-Hard-Auto questions. We find that, even after changing all of the questions, the Pearson’s correlation with the original judgments is .94, which strongly suggests that the content of the questions plays little role in the judgments.
5. **(Factor analysis.)** Following the requests of reviewers, we have added a factor analysis of Arena-Hard judgments (for a single judge) to Appendix K. We observe three factors which account for roughly 64% of the variance, which we term quality, accuracy and brevity. Troublingly, safety has very low loadings on all factors and a very low communality (0.04), suggesting it's measuring something almost entirely independent of the other metrics.

## Common Questions

**CQ1.** Why is SOS-Bench an important contribution, considering that HELM and Open LLM Leaderboard already exist?

**CA1.** While existing premier static benchmarks such as HELM Core (itself a meta-benchmark) and LiveBench are very important, and inspired aspects of our work, they are not designed as alignment benchmarks in the way that Arena-Hard-Auto is. The Open LLM Leaderboard has issues with saturation on some of its tasks, and only measures world knowledge. LiveBench has an implicit dependence on instruction following because all of its questions are open-ended, making it difficult to precisely assess the performance of smaller and older LLMs which tend to be used in academic papers. The HELM family is complex, with many non-intersecting metrics, models and datasets; important factors in model alignment, such as safety (bias and toxicity in HELM) are not covered for all models, and instruction following is mostly evaluated implicitly. Finally, it is not obvious how one would compare alignment methods (E.G., DPO vs PPO) using HELM. Therefore, we believe that SOS-Bench has the potential to be high-impact; it combines existing benchmarks in a novel way that makes it easy to measure progress on salient factors in alignment. We provide code to reproduce our benchmark exactly (and we will provide all relevant model weights once we are de-anonymized), which will make it easy to compare future work to our baselines. We also note that meta-benchmarks have served as a key contribution for multiple recent publications: https://arxiv.org/abs/2308.11838 (ICLR24), https://arxiv.org/pdf/2306.09468 (ICLR24).

---

> ### Author Response · Authors · 2024-12-01
>
> We thank the reviewers for their feedback thus far. We wish to politely remind the reviewers that we are approaching the deadline for author-reviewer discussion phase -- at this time, we would appreciate a final response to our comments, and if warranted, a re-evaluation of the review scores. Thank you all for your hard work!
>
> -- The Authors

---

### Meta-Review · Area_Chair_ejfp · 2024-12-23

**Metareview:**

The paper investigates a well motivated problem: do LLM-judge preferences translate to progress on other, more concrete metrics for alignment, and if not, why not? In particular, the work finds that LLM-judges have implicit biases, and prioritize stylistic preferences over other more important substances like factuality and safety. They then present a new alignment benchmark, SOS-bench, with the goal of evaluating truthfully the progress on helpful, honest, and harmless principles. Experiments show that data scaling in SFT and improving prompt diversity are important predictors of improved alignment.

All reviewers rated positively towards the work. More importantly, through the helpful and productive discussion period, we think the work is improved substantially with the amount of ablation studies added.

**Additional Comments On Reviewer Discussion:**

The discussion overall is very healthy and productive. The author was able to convince one reviewer to update their score, and generally improved their paper with valuable suggestions from everyone.

---

### Decision · Program_Chairs · 2025-01-22

Accept (Poster)